# Quality-of-Life Determinants in People with Diabetes Mellitus in Europe

**DOI:** 10.3390/ijerph18136929

**Published:** 2021-06-28

**Authors:** Álvaro Fuentes-Merlos, Domingo Orozco-Beltrán, Jose A. Quesada Rico, Raul Reina

**Affiliations:** 1Department of Primary Health Care, San Juan de Alicante University Hospital, 03550 San Juan de Alicante, Spain; alvaro.fuentes@goumh.umh.es; 2Department of Clinical Medicine, Miguel Hernández University, 03550 San Juan de Alicante, Spain; dorozco@umh.es; 3Department of Sport Sciences, Sport Research Centre, Miguel Hernández University, 03202 Elche, Spain; rreina@umh.es

**Keywords:** diabetes mellitus, health surveys, quality of life, Europe

## Abstract

This study aims to analyze self-perceived health and lifestyles in the European Union Member States Iceland, Norway, and the United Kingdom, examining associations with diabetes prevalence; and to identify the demographic, economic and health variables associated with diabetes in this population. We performed a cross-sectional study of 312,172 people aged 15 years and over (150,656 men and 161,516 women), using data collected from the European Health Interview Survey (EHIS). The EHIS includes questions on the health status and health determinants of the adult population, as well as health care use and accessibility. To estimate the magnitudes of the associations with diabetes prevalence, we fitted multivariate logistic models. The EHIS data revealed a prevalence of diabetes in Europe of 6.5% (*n* = 17,029). Diabetes was associated with being physically inactive (OR 1.14; 95% CI 1.02–1.28), obese (OR 2.75; 95% CI 2.60–2.90), male (OR 1.46; 95% CI 1.40–1.53) and 65–74 years old (OR 3.47; 95% CI 3.09–3.89); and having long-standing health problems (OR 7.39; 95% CI, 6.85–7.97). These results were consistent in the bivariate and multivariate analyses, with an area under the receiver operating characteristic curve of 0.87 (95% CI 0.87–0.88). In a large European health survey, diabetes was clearly associated with a poorer perceived quality of life, physical inactivity, obesity, and other comorbidities, as well as non-modifiable factors such as older age and male sex.

## 1. Introduction

Improvements in health interventions are defined in terms of their efficacy and, less frequently, their efficiency [1]. Innovative health strategies such as the Triple Aim framework adopt a multifactorial approach to health intervention assessment, analyzing efficacy and cost (efficiency), while also incorporating patients’ perspectives. Quality of life forms part of the latter factor but is rarely included in health evaluations. Taking patients’ experiences into account is particularly important in chronic diseases. In addition, the ageing of the European population calls for new strategies to assess the association of lifestyles and various socio-demographic variables with chronic diseases [2,3].

Diabetes is one of the leading causes of death in Western countries. Patients can suffer acute complications (e.g., hypoglycemia or ketoacidosis) as well as long-term microvascular complications (e.g., retinopathy, neuropathy, or nephropathy) or macrovascular disease (e.g., stroke or heart disease), all of which result in greater disability (i.e., limitation of activities of daily living) and an enormous financial burden for European health systems [2].

According to recent estimates by the International Diabetes Federation, in 2019, there were 463 million people living with diabetes worldwide (9.3% of the population aged between 18 and 99 years), half of them undiagnosed. This figure is set to reach 10.2% (578 million) by 2030 and 10.9% (700 million) by 2045 [3]. In Europe alone, an estimated 58 million adults have diabetes, and this number is projected to increase to 66.7 million by 2045 (from 8.8% to 10.2%) [4,5].

We can assess the impact of diabetes on patients by analyzing health-related quality of life (HRQoL), which is defined as a multidimensional concept including domains related to physical, mental, emotional, and social functioning [6]. Data submitted by patients are fundamental for evaluating the impact of this type of chronic disease, as well as the effectiveness of treatments and health policy planning. According to one systematic review [5], health–economic evaluations of new therapies must consider the effects and side-effects of each intervention on HRQoL (i.e., self-management, glycemic control, and obesity).

The literature shows that lifestyle (e.g., diet, smoking, and physical activity) and psychosocial factors (e.g., depression, low socioeconomic status, and older age) may be predictors of diabetes [7,8,9]. Daily moderate physical activity, such as brisk walking, improves health and is inversely associated with diabetes, obesity, cancer, cardiovascular disease, and premature mortality [10,11,12,13]. A healthy diet (e.g., high in fiber and low in saturated and trans fats) is also crucial for preventing chronic diseases and improving HRQoL [13,14]. In addition, depression is considered to be a risk factor for diabetes and its complications [15,16], with a higher risk of developing the disease observed among adults with depression [17]. As a result, encouraging lifestyle changes to improve dietary habits and increase physical activity forms the cornerstone of diabetes prevention strategies in Europe [9,18].

The fact that low educational attainment and residence in economically depressed areas are also risk factors for diabetes supports the view that health education is crucial in diabetes management. Self-management—a strategy that emphasizes patients’ responsibility in the treatment of their disease—could help to reduce the use of health care services and thus, reduce health costs [19]. For people with diabetes, self-management encompasses a wide range of activities, including daily administration of oral medication and/or insulin, blood glucose monitoring, healthy eating, and regular physical activity. Participation in diabetes self-management education programs is associated with a healthier and more active lifestyle [20], as well as improved HRQoL [21]. Promoting self-management in people with chronic diseases is, therefore, an important strategy for improving HRQoL and health system sustainability, and health care providers must focus on optimizing the scope and effectiveness of self-management support in people with diabetes [20].

In the literature, we found no previous studies that used data from the European Health Interview Survey (EHIS) to analyze lifestyles, efficacy of health interventions and use of health care services in people with diabetes living in Europe. In this study, we aim to analyze self-perceived health and lifestyles in the European Union Member States Iceland, Norway, and the United Kingdom, examining associations with diabetes prevalence; and to identify the demographic, economic and health variables associated with diabetes in this population.

## 2. Materials and Methods

### 2.1. Sample

This is a cross-sectional study that aims to identify factors related to diabetes in the European population. The data used for this purpose had been collected as part of the EHIS and were provided by the EU Statistical Office (Eurostat) [22]. The EHIS was implemented in 2013 in Belgium and the United Kingdom; in 2014, in Bulgaria, Czech Republic, Estonia, Greece, Spain, France, Croatia, Italy, Cyprus, Latvia, Lithuania, Luxembourg, Hungary, Malta, Netherlands, Austria, Poland, Portugal, Romania, Slovenia, Slovakia, Finland, and Sweden; and in 2015, in Denmark, Germany, Ireland, Italy, Iceland, and Norway. The statistical population for the survey includes all people aged 15 years and over living in private households and residing in any of the above countries. The exclusion criteria for our study were incomplete or erroneous responses for any of the study variables.

### 2.2. Study Variables

The explanatory variables included in this study were sex, age, country of residence, degree of urbanization, legal marital status, hours of recreational physical activity, consumption of fruit and vegetables, smoking, exposure to smoke, alcohol consumption, self-perceived general health, suffering from different diseases in the past 12 months, body mass index, physical and sensory functional limitations, intensity of bodily pain, severity of depression (Patient Health Questionnaire (PHQ-8)) [22], long-standing health problems (at least 6 months), last time of vaccination against flu, educational attainment, labor status, number of people to count on in the event of serious personal problems, number of people living in the household, type of household, and socioeconomic status. The primary variable of the study was the presence or absence of type 1 and type 2 diabetes.

### 2.3. Statistical Analysis

We performed a descriptive analysis by calculating the frequencies of all qualitative variables; and the minimum, maximum, mean and standard deviation of all quantitative variables. We analyzed the factors associated with diabetes prevalence using contingency tables, applying the Chi-Square test for the qualitative variable and the Student *t* test for the quantitative variables. To estimate the magnitudes of the associations, we fitted multivariate logistic models. Odds ratios (ORs) were calculated with the corresponding 95% confidence intervals (95% CI). Variables were selected by a stepwise procedure based on the Akaike information criterion. We calculated the likelihood ratio test (LRT) statistic to measure goodness of fit, and the area under the receiver operating characteristic (ROC) curve.

Five countries (Spain, Belgium, France, Italy, The Netherlands) did not provide data for hours of recreational physical activity, physical and sensory functional limitations, alcohol consumption, or severity of depression. To obtain the largest possible amount of information from the survey, and given the importance of these variables for the objectives of this study, we used the responses of all countries for the descriptive and univariate analyses, and only the responses of countries with valid data for all variables in the multivariate analysis.

To obtain representative estimates of the European population, we took into account the complex sample design, using, as a weighting factor, the raising factor of the survey divided by its mean in each country, obtaining weights centered on the means [23,24]. For the analyses, we used SPSS (version 26, SPSS Inc., Chicago, IL, USA) and R Statistical software (version 4.0.2, R Core Team, R Foundation for Statistical Computing, Vienna, Austria).

## 3. Results

We analyzed 312,172 participants aged 15 years and older, of whom 51.7% were women and 48.3% men. By age group, those aged 74 years or older represented 9.8% of the population, while those under 40 years of age accounted for 37.6% of the total. In relation to health status, 7.8% of survey respondents reported having bad or very bad self-perceived general health, 3.8% had chronic obstructive pulmonary disease (COPD), 20.9% had hypertension, and 6.5% had diabetes (the primary variable of our study).

In total, 15.8% had some physical limitation and 23.5% reported moderate, severe, or very severe bodily pain. The PHQ-8 findings showed that 11.8% of participants had mild depression and 4.9% had moderate or severe depression (Appendix A). Regarding health variables, 16% of respondents had received a flu vaccine in the previous year. Half (49.9%) were in work and 22.8% were retired. Concerning lifestyle habits, 61.7% of participants reported doing no recreational physical activity, and 53.5% and 49.2% ate fruit and vegetables every day, respectively. Meanwhile, 19.3% smoked every day and 49% drank alcohol more than once weekly.

Table 1 shows the results of the univariate analysis with the percentages of health status variables in the diabetic and non-diabetic population. Bad or very bad self-perceived health was significantly higher in people with diabetes (26.5%) than in those without (6.4%). Diabetes was also associated with greater comorbidity: COPD (10.4% vs. 3.3%), hypertension (58.8% vs. 18.3%), coronary heart disease or angina pectoris (13.1% vs. 3%) and depression (13.5% vs. 6.6%). Moderate, severe, or very severe bodily pain (43.7% vs. 22.1%) and the presence of a long-standing health problem (90.9% vs. 39.9%) were more prevalent in the diabetic population. Appendix A includes all the variables.

Table 2 presents the health status variables included in the adjusted multivariate logistic regression model for presence of diabetes. The survey responses from Spain, Belgium, France, Italy, and the Netherlands were not included owing to invalid data in some variables. After adjusting for confounders, diabetes was associated with poorer self-perceived general health, with odds ratios for diabetes of 3.42 and 3.38 in people with bad or very bad perceived health compared with those who reported very good perceived health. Similarly, diabetes was much more frequent in people with long-standing health problems versus those without (OR 7.39; 95% CI 6.85–7.97). Other factors associated with diabetes prevalence were male sex (OR 1.46; 95% CI 1.40–1.53), older age and higher number of comorbidities, such as myocardial infarction (OR 1.25; 95% CI 1.15‒1.36), hypertension (OR 1.64; 95% CI 1.58‒1.71), cirrhosis (OR 1.97; 95% CI 1.68‒2.32), kidney problems (OR 1.50; 95% CI 1.40‒1.61) and obesity (OR 2.75; 95% CI 2.60‒2.90). The presence of diabetes was clearly related to a lack of physical activity (OR 1.14; 95% CI 1.02–1.28) and severe physical limitation (OR 1.19; 95% CI 1.12–1.27). In the multivariate analysis, no association was found between depression and diabetes, leading us to conclude that the higher prevalence of depression observed in people with diabetes in the univariate analysis (13.5% vs. 6.6%) must have been conditioned by other factors such as older age. In relation to socioeconomic variables, diabetes was associated with lower educational attainment (OR 1.29; 95% CI 1.19–1.41) and domestic labor status (OR 1.28; 95% CI 1.15–1.43). In contrast, lower consumption of fruits was associated with a lower prevalence of diabetes.

The model fit the data well, with a final sample size of 232,386 participants, including 17,029 people with diabetes (7.33%), an area under the ROC curve of 0.87 and LRT statistic of 32,071.1 (*p* value < 0.001). Appendix A presents the complete model, adjusted for age, sex, BMI, country of residence, educational attainment, labor status, consumption of fruit and vegetables, household income, accidents at home and flu vaccination. In addition, Appendix A include all the model indicators.

## 4. Discussion

The results of this study confirm the relationship between diabetes prevalence and lifestyle, health determinants and psychosocial variables in Europe. In our European population, 6.5% of people had diabetes, and the prevalence was higher in southern compared with northern countries. While 6.5% is lower than the 9.3% worldwide prevalence, our findings are consistent with those of previous studies and reflect the huge and growing burden of diabetes in Europe, which varies considerably between countries and income groups [2,3,4].

Regarding sex and age, the data reflect a positive association between diabetes and being a man aged 65 and older. Although lifetime risk of diabetes is similar in men and women, there are important differences between the sexes in terms of onset age, detection, and disease burden [25]. For example, middle-aged men have a higher prevalence of type 2 diabetes than women of the same age, while older women have a higher prevalence than older men [26,27]. Aging is known to influence the risk of type 2 diabetes, which accounts for 90% of diabetes cases worldwide [4].

The main driver of diabetes costs is the treatment of associated complications. The European diabetic population of this study had a higher prevalence of microvascular and macrovascular complications, including loss of visual acuity, kidney problems, coronary heart disease, myocardial infarction, hypertension, and cirrhosis compared to people without diabetes. The association between diabetes and these comorbidities is reflected in the specialized literature [5,6,7,8,18,21,28]. All of these factors have a direct impact on HRQoL. Indeed, our results are consistent with previously published evidence [5,6,14,15,16], in that they suggest an association between diabetes and severe physical limitation, very bad self-perceived health and the presence of a long-standing health problem.

Low educational attainment, low household income and greater use of health services (i.e., number of visits to a family doctor in the past four weeks or number of inpatient hospital admissions in the past 12 months) were significantly associated with diabetes in our European population. These findings are in line with the results of previous studies, which demonstrate a link between low income and educational attainment with diabetes [8,29], and highlight the importance of education for the prevention and treatment of the disease [20].

Concerning lifestyle variables (i.e., physical activity and fruit and vegetable consumption), obesity and lack of recreational physical activity were also significantly associated with diabetes. These results are in line with the existing evidence suggesting that lifestyle and physical activity should form the cornerstone of diabetes prevention and treatment [12,13]. According to recent studies [12,30,31], lifestyle interventions (e.g., physical activity, improved diet, or weight loss) can reduce diabetes incidence by up to 58%. Furthermore, there is abundant epidemiological and clinical evidence that shows an association between physical activity and reduced risk of developing certain diseases such as diabetes, obesity, cardiovascular diseases, and depression by up to 60% [30,31,32]. Although we adopted a cross-sectional design and were therefore unable to establish causality, our results are consistent with those of other studies that demonstrate how physical activity and a healthy lifestyle confer considerable protection against diabetes [10,11,12,13,32].

Although healthy eating includes several aspects such as reducing the consumption of sugary drinks and low-fiber foods, the EHIS 2013–2015 study provides data on the consumption of fruit and vegetables only. In contrast to the international literature, we found a higher frequency of fruit and vegetable consumption in participants with diabetes compared to those without. This could be a result of doctors providing their diabetic patients with dietary recommendations, leading to a greater awareness of healthy eating as a key factor of diabetes management [20]. On the other hand, eating foods rich in soluble fibers, such as vegetables and fruit, does not appear to significantly reduce diabetes risk [33], which could explain the apparent lack of protective effect in our population.

It is worth mentioning that our study is population-based, with a very large sample that is representative of the whole European population. We used the raising factors of each surveyed country, meaning the estimations adequately reflect population parameters. Moreover, the variables were designed for a multilevel health questionnaire and have been validated in large-scale health interview surveys in Europe, where the EHIS has been mandatory in all Member States and in Iceland, Norway, and the United Kingdom since 2013 [34]. Random population selection and analysis of the harmonized EHIS data ensure quality and comparability of the health information in the countries involved, enabling better testing of public policies. However, future research could include between-country comparisons based on more detailed information to better understand the effect of national levels of development, healthcare systems, or other socio-economic factors on diabetes prevalence.

Some limitations of this study should be mentioned. Firstly, the EHIS questionnaire only provides cross-sectional data, and we were, therefore, unable to evaluate longitudinal trends and causal relationships. All the data and variables examined in our study are based on self-reports and could therefore be affected by recall bias and social desirability bias, though in the design and validation of the EHIS, efforts are made to minimize the effects of non-response and self-reported biases [32]. Diabetes itself is a cause of low self-perceived health, and many of the poorer health indicators in the diabetic population of our study may in fact be due to the disease rather than being risk factors for the disease. Nevertheless, the findings of this study could help us to better understand the true risk factors for diabetes, as well as the potential consequences of this disease on health and quality of life in a large sample of European citizens. Secondly, the questionnaire excludes gestational diabetes from the definition of diabetes but does not distinguish between type 1 and type 2 diabetes. However, given the large sample size and the high prevalence of type 2 diabetes in Europe [2,3,4], we can assume that the great majority of diabetic participants had type 2 diabetes. Thirdly, the statistical tests used tend to give significant results because of the large sample size; for this reason, we tried to evaluate clinical significance as well as statistical significance, taking effect size into account. Another limitation involves the potential under-reporting of diabetes. Although comorbidities such as myocardial infarction and coronary disease were associated with diabetes prevalence in our study, there is evidence to show that in people with coronary atherosclerosis, diabetes is often undiagnosed [35,36]. We nevertheless assume that self-reported diabetes in the EHIS is based on previous clinical diagnosis. Finally, we should stress that five countries (Spain, Belgium, France, Italy, and the Netherlands) did not provide information about certain variables analyzed in this study. Because these variables were included in the multivariate logistic model, the resulting estimations are not representative of those countries.

## 5. Conclusions

This study confirms the association between lifestyle and diabetes prevalence in Europe, and the influence of demographic variables such as educational attainment, age, and sex on these associations. In view of the ageing populations and increasing socioeconomic and demographic diversity in multinational and multicultural regions such as Europe, effective prevention of diabetes requires multidimensional public health programs that incorporate patients’ perspectives (i.e., physical, emotional, and social functioning), lifestyles and socioeconomic status (education and income). This innovative approach—based not only on improving life expectancy and socioeconomic indicators, but also on the experience of a population that increasingly requires HRQoL services—should focus on promoting healthy lifestyles and education for self-management of chronic diseases. Future studies could assess the impact of such an approach on diabetes care, a key component of the economic burden on healthcare systems. Our results suggest that targeted lifestyle interventions (e.g., educational self-management workshops or promotion of physical activity and weight loss) in specific sectors of the population (e.g., people with obesity, aged 55 and over or with comorbidities) could improve quality of life in people with diabetes and enable a more efficient use of health services in Europe.

## Figures and Tables

**Table 1 ijerph-18-06929-t001:** Health status variables in the diabetic and non-diabetic population.

Health Status Variables	No Diabetes		Diabetes		
	*n*	%	*n*	%	*p* Value
Self-perceived general health					
Very good	74,216	25,4	613	3.0	<0.001
Good	130,289	44.6	4884	24.0	
Fair	58,745	20.1	8894	43.7	
Bad	15,020	5.1	4194	20.6	
Very bad	3707	1.3	1203	5.9	
Don’t know/refusal	9844	3.4	562	2.8	
Long-standing health problem (duration ≥ 6 months)					
No	17,1603	58.8	1612	7.9	<0.001
Yes	116,518	39.9	18,499	90.9	
Don’t know/refusal	3700	1.3	240	1.2	
COPD in the past 12 months					
No	281,369	96.4	17,805	87.5	<0.001
Yes	9659	3.3	2108	10.4	
Don’t know/refusal	794	0.3	437	2.1	
MI/chronic consequences of MI *					
No	287,497	98.5	18,520	91.0	<0.001
Yes	3521	1.2	1374	6.8	
Don’t know/refusal	804	0.3	457	2.2	
Coronary heart disease/angina pectoris *					
No	282,345	96.8	17,257	84.8	<0.001
Yes	8616	3.0	2657	13.1	
Don’t know/refusal	861	0.3	436	2.1	
Hypertension *					
No	238,080	81.6	8237	40.5	<0.001
Yes	53,366	18.3	11,973	58.8	
Don’t know/refusal	376	0.1	140	0.7	
Stroke/chronic consequences of stroke *					
No	287,724	98.6	18,902	92.9	<0.001
Yes	3257	1.1	982	4.8	
Don’t know/refusal	841	0.3	466	2.3	
Cirrhosis of the liver *					
No	290,731	99.6	19,570	96.2	<0.001
Yes	869	0.3	353	1.7	
Don’t know/refusal	222	0.1	427	2.1	
Kidney problems *					
No	284,834	97.6	18,040	88.6	<0.001
Yes	6726	2.3	1912	9.4	
Don’t know/refusal	262	0.1	398	2.0	
Depression *					
No	272,173	93.3	17,226	84.6	<0.001
Yes	19,134	6.6	2738	13.5	
Don’t know/refusal	516	0.2	386	1.9	
Wearing glasses or contact lenses					
No	128,205	43.9	4067	20.0	<0.001
Yes	163,187	55.9	16,241	79.8	
Don’t know/refusal	430	0.1	42	0.2	
Physical limitation					
No difficulty	236,831	81.2	10,362	50.9	<0.001
Moderate difficulty	24,361	8.3	4572	22.5	
Severe difficulty	15,626	5.4	4746	23.3	
Don’t know/refusal	15,005	5.1	670	3.3	
Intensity of bodily pain during the past 4 weeks					
None	141,928	48.6	5682	27.9	<0.001
Very mild	37,087	12.7	2207	10.8	
Mild	39,509	13.5	3062	15.0	
Moderate	41,170	14.1	4916	24.2	
Severe	18,307	6.3	2998	14.7	
Very severe	4966	1.7	985	4.8	
Don’t know/refusal	8854	3.0	500	2.5	
Depression severity (PHQ-8)					
None/minimal (0–4)	195,611	67.0	10,866	53.4	<0.001
Mild (5–9)	33,189	11.4	3599	17.7	
Moderate (10–14)	8451	2.9	1378	6.8	
Severe (15–24)	4807	1.6	923	4.5	
Don’t know/refusal	49,764	17.1	3584	17.6	

* in the past 12 months. COPD: chronic obstructive pulmonary disease; MI: myocardial infarction; PHQ-8: Patient Health Questionnaire.

**Table 2 ijerph-18-06929-t002:** Multiple logistic regression model for presence of diabetes.

Health Status Variables	OR	95% CI	*p* Value
Self-perceived general health			
Very good	1		
Good	1.61	(1.45–1.80)	<0.001
Fair	2.52	(2.25–2.81)	<0.001
Bad	3.42	(3.03–3.87)	<0.001
Very bad	3.38	(2.92–3.91)	<0.001
Don’t know/refusal	1.88	(1.35–2.60)	<0.001
Long-standing health problem (duration ≥ 6 months)			
No	1		
Yes	7.39	(6.85–7.97)	<0.001
Educational attainment			
Tertiary education; bachelor, master or doctoral level	1		
Tertiary education; short cycle	0.98	(0.90–1.08)	0.752
Secondary education	1.14	(1.07–1.22)	<0.001
Primary education	1.29	(1.19–1.41)	<0.001
Don’t know/refusal	1.15	(0.88–1.50)	0.306
Labor status			
In work	1		
Unemployed	1.09	(0.90–1.21)	0.088
Studying	0.85	(0.70–1.02)	0.088
Retired	1.21	(1.13–1.30)	<0.001
Domestic tasks	1.28	(1.15–1.43)	<0.001
Other inactive	1.14	(1.05–1.24)	0.001
Don’t know/refusal	1.20	(0.97–1.48)	0.094
Body mass index			
Normal	1		
Overweight	1.57	(1.49–1.65)	<0.001
Obese	2.75	(2.60–2.90)	<0.001
Don’t know/refusal	1.73	(1.53–1.95)	<0.001
Hours of recreational physical activity			
> 7 h a week	1		
3–7 h a week	1.01	(0.89–1.15)	0.821
1–3 h a week	1.04	(0.93–1.17)	0.485
Don’t know/refusal	1.14	(1.02–1.28)	0.020
Frequency of eating fruit			
Once or more a day	1		
4–6 times a week	0.90	(0.85–0.95)	<0.001
1–3 times a week	0.86	(0.814–0.91)	<0.001
<once a week	0.88	(0.81–0.95)	0.001
Never	0.81	(0.68–0.97)	0.025
Don’t know/refusal	0.74	(0.53–1.03)	0.079
Household income			
Below 1st quintile	1		
Between 1st and 2nd quintile	1.09	(1.03–1.16)	0.002
Between 2nd and 3rd quintile	1.02	(0.97–1.09)	0.412
Between 3rd and 4th quintile	1.01	(0.95–1.08)	0.744
Between 4th and 5th quintile	1.00	(0.93–1.08)	0.918
Don’t know/refusal	0.97	(0.88–1.07)	0.554
MI or chronic consequences of MI *			
No	1		
Yes	1.25	(1.15–1.36)	<0.001
Don’t know/refusal	1.51	(1.06–2.16)	0.023
Coronary heart disease or angina pectoris *			
No	1		
Yes	1.09	(1.03–1.16)	0.003
Don’t know/refusal	1.12	(0.82–1.53)	0.462
Hypertension *			
No	1		
Yes	1.64	(1.58–1.71)	<0.001
Don’t know/refusal	1.64	(1.13–2.39)	0.009
Cirrhosis of the liver *			
No	1		
Yes	1.97	(1.68–2.32)	<0.001
Don’t know/refusal	2.94	(2.03–4.28)	<0.001
Kidney problems *			
No	1		
Yes	1.50	(1.40–1.61)	<0.001
Don’t know/refusal	1.565	(1.06–2.30)	0.022
Wearing glasses or contact lenses			
No	1		
Yes	1.19	(1.13–1.25)	<0.001
Don’t know/refusal	0.78	(0.51–1.21)	0.270
Physical limitation			
No difficulty	1		
Moderate difficulty	1.10	(1.04–1.15)	<0.001
Severe difficulty	1.19	(1.12–1.27)	<0.001
Don’t know/refusal	1.16	(0.83–1.62)	0.372
Intensity of bodily pain during the past 4 weeks			
None	1		
Very mild	0.87	(0.83–0.93)	<0.001
Mild	0.76	(0.71–0.81)	<0.001
Moderate	0.74	(0.70–0.79)	<0.001
Severe	0.70	(0.66–0.76)	<0.001
Very severe	0.77	(0.69–0.86)	<0.001
Don’t know/refusal	0.73	(0.54–0.98)	0.039

* in the last 12 months. OR: Odds Ratio; OR adjusted for age, sex, body mass index, country of residence, educational attainment, labor status, fruit and vegetable consumption, income, home accidents, and last time of vaccination against flu. Model indicators: total population 232,386; population with diabetes 17,029; likelihood ratio test (LRT) 32,071.1; *p* value < 0.001; area under the receiver operating characteristic curve (AUC) 0.87; 95% CI 0.87–0.88.

## Data Availability

All original data are included in the EHIS database provided by Eurostat.

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
