# Peer review of "Quality-of-Life Determinants in People with Diabetes Mellitus in Europe"

_ijerph, 2021, doi:10.3390/ijerph18136929_

Round 1

Reviewer 1 Report

While I feel that most people already associated diabetes with poorer perceived quality of life, this research can be used as scientific evidence to reinforce this association. Having said this, I feel that it might be necessary to add how future researchers can be built from this finding, like how can we improve the perceived quality of life.

Your abstract is good. It is unnecessary to write Objective: Research Design and Methods, and Result. Your writing is already self-explanatory.

Your introduction and conclusion need improvement. I have to read until the end of the introduction before I know what this paper is about. That piece of information should come as early as possible. Your conclusion is very short, and I do not get much information out of it. Also, did you encounter any difficulty while performing this research (I see that some countries do not provide certain information)? How would you avoid this in the future, and are there any further research recommendations?

Regarding your results, are there any differences between countries? Why you do not provide this information. I strongly believe that different development levels, healthcare systems and other socio-economic factors play a big part. Is there any reason behind this? Could you address this issue? – Your research has more potential in providing much more information. It is rather short.

Check your writing style. The overall grammar is good. However, there are some inconsistence and missing punctuation here and there.

Author Response

Dear reviewer.

Please, see the responses to your queries in the attached file.

Sincerely.

Reviewer 2 Report

Good job, there is a high sample in this study, which allows to explore variables in many different ways. However, some changes are required.

Although it appears in the title (just once in the whole paper), I would clarify in the background it is Diabetes Mellitus (DM) what you are referring to and would specify if the data available is about type 1, type 2 DM or both. 

For better scheming, I recommend to switch 2nd and 3rd paragraph (up to "an enormous financial burden for European health systems (5)"), so DM would be introduced before epidemiology data in this way.  

In the results, it would be avisable to highlight there is an opposite trend in age: as age increases in non-diabetic subjects, the sample diminishes; while in subjects with DM, there is an increase of the sample as age also increases. This leads to some considerations regarding size effect across age groups: statistics and signification in subjects >74 years old (24,817 non-diabetic vs 5,705 with DM) might be different than in subjects <40 years old (116,201 vs 1,150 respectively).

As there is a distinction made between lifestyle habits variables and health status variables in the results, discussion should appear organised similarly. In addition, that division of variables could be clearer if noted in the tables as that. 

As your aim is "to analyze self-perceived health and lifestyles, examining associations with diabetes prevalence; and to identify the demographic, economic and health variables associated with diabetes" conclusions go way further, up to the extent you are assuming that current interventions are not effective (which is not aimed). I encourage you to rather approach this in the discussion.

Author Response

(The authors gave the same response as above.)

Reviewer 3 Report

The goal of this study - to determine how quality of life influences risk of a chronic disease like diabetes - is a worthy one but it only partly achieved in this study.  Diabetes causes visual problems, kidney and cardiovascular disease, pain and neuropathies and is all by itself a cause of a perception of poor health, because diabetic do have poor health. 

Diabetes is known to be caused by poor diet and inadequate exercise, so it is not surprising the diabetic show these signs.  Thus most of the parameters shown to be higher among diabetes may be due to the disease, not a risk factor for causing the disease.  This must be fully discussed, and it is not. 

The conclusion that these factors led to diabetes is not established by most of these results.  There is merit in this study, which confirms much of what we know about diabetes, but the conclusions are presented in an uncritical fashion because of not adequately considering the results of having diabetes. 

By the nature of the design of the study you did not show that obesity and lack of exercise "caused" diabetes, only that people with diabetes were more obese and did not exercise.   

The discussion and conclusions must be significantly revised so as to acknowledge that many of the findings may be due to the diabetes, rather than being risk factors for the diabetes.  

Author Response

(The authors gave the same response as above.)

Reviewer 4 Report

In the present study Dr Álvaro Fuentes-Merlos et al aimed to analyze the association of the diabetes prevalence with self-perceived health and lifestyles in several European Union Member States, using data collected for the European Health Interview Survey (EHIS). As expected, the find that diabetes is clearly associated with well known risk factors such as physical inactivity, obesity, older age and male sex. Interestingly, they also report a significant association with other covarietes such as a poorer perceived quality of life.

The manuscript is nice and well written. I have only minor points:

- older age is one of the many study variables significantly associated with diabetes prevalence. However, as stated in Table 2, 65-74 y has an higher OR than age >74 y. This result is quite surprising because it well knows that diabetes prevalence increases in very old individuals. Looking at study methods, authors declare that participants aged 74 years or over represented only the 9.8% of the study population while in every european country, the majority of patients with diabetes aged over 75 y. Authors have to comment this point including among limitation.

- authors report that MI and/or Coronary heart disease are significantly associated with diabetes diagnosis. However, It is also known that in subjects affected by atherosclerotic vascular diseases, the presence of diabetes is often undiagnosed. Please include a comment regarding the possible magnitude of underdiagnosis, by reffering the following publications: 1) Adiponectin isoforms are not associated with the severity of coronary atherosclerosis but with undiagnosed diabetes in patients affected by stable CAD. doi: 10.1016/j.numecd.2007.12.001, 200. Nutr Metab Cardiovasc Dis and 2) Occult impaired glucose regulation in patients with atherosclerosis is associated to the number of affected vascular districts and inflammation. Atherosclerosis. 2010. doi: 10.1016/j.atherosclerosis.2010.05.017.

Author Response

(The authors gave the same response as above.)
